# Salicylic Acid Protects Sweet Potato Seedlings from Drought Stress by Mediating Abscisic Acid-Related Gene Expression and Enhancing the Antioxidant Defense System

**DOI:** 10.3390/ijms232314819

**Published:** 2022-11-26

**Authors:** Chongping Huang, Junlin Liao, Wenjie Huang, Nannan Qin

**Affiliations:** 1Department of Agronomy, College of Agriculture and Biotechnology, Zhejiang University, 866 Yu-Hang-Tang Road, Hangzhou 310058, China; 2Hainan Institute of Zhejiang University, Sanya 572025, China; 3Agricultural Experiment Station of Zhejiang University, 866 Yu-Hang-Tang Road, Hangzhou 310058, China

**Keywords:** sweet potato, drought stress, salicylic acid, photosynthetic capability, antioxidant enzymes, *NCED-like3* expression

## Abstract

China has the largest sweet potato planting area worldwide, as well as the highest yield per unit area and total yield. Drought is the most frequently encountered environmental stress during the sweet potato growing season. In this study, we investigated salicylic acid (SA)-mediated defense mechanisms under drought conditions in two sweet potato varieties, Zheshu 77 and Zheshu 13. Drought stress decreased growth traits, photosynthetic pigments and relative water contents, as well as the photosynthetic capability parameters net photosynthetic rate, stomatal conductance and transpiration rate, whereas it increased reactive oxygen species production, as well as malondialdehyde and abscisic acid contents. The application of SA to drought-stressed plants reduced oxidative damage by triggering the modulation of antioxidant enzyme activities and the maintenance of optimized osmotic environments in vivo in the two sweet potato varieties. After SA solution applications, *NCED-like3* expression was downregulated and the abscisic acid contents of drought-stressed plants decreased, promoting photosynthesis and plant growth. Thus, foliar spraying an appropriate dose of SA, 2.00–4.00 mg·L^−1^, on drought-stressed sweet potato varieties may induce resistance in field conditions, thereby increasing growth and crop yield in the face of increasingly frequent drought conditions.

## 1. Introduction

Drought is the most frequent environmental problem in crop production worldwide. Owing to climate change, seasonal and regional droughts are showing trends of increasing [1]. Plants have evolved complex morphological, physiological and biochemical mechanisms to cope with drought stress [2]. When the soil moisture is insufficient, the stomata often close partially or completely to decrease the transpiration rate, which reduces water loss and CO_2_ entry, resulting in a photosynthetic rate decrease. Deficiencies in CO_2_ and H_2_O lead to excessive energy production by electron transmission during photosynthesis, which further induces the overgeneration of various reactive oxygen species (ROS) [3]. Plants have developed an antioxidant system, including the antioxidant enzymes superoxide dismutase (SOD), catalase (CAT), peroxidase (POD) and ascorbate peroxidase, as well as the low-molecular antioxidant compounds ascorbate, glutathione, α-tocopherol, β-carotene and flavonoids, to scavenge high levels of ROS [4,5]. To maintain water balance in vivo, plants produce soluble sugars and proteins that increase the osmotic potential. Furthermore, drought induces changes in the expression levels of related genes, resulting in the inhibition of normal and stress-specific protein synthesis [6]. However, there are limited reports regarding the physiological responses of sweet potato, *Ipomoea batatas*, to the foliar spraying of SA under drought conditions.

As an important plant hormone, SA plays a key regulatory role in plants growing under environmentally stressed conditions. Applying an appropriate SA dose results in plant tolerance to drought, salt, cold and heat, as well as to heavy metal toxicity stress, and it enhances systemic acquired resistance when plants are infected by pathogens [7]. The exogenous application of SA can alleviate the impacts of chilling stress on *Dendrobium. officinale* seedlings by protecting the chloroplast membrane, including PS II D1 protein, and it enhances the antioxidant capacity of plants [8]. In a high temperature environment, SA treatments of ornamental pepper crops increase the germination rate and germination potential, and they reduce oxidative damage to seeds [9]. Additionally, SA remarkably increases the total polyphenolic content and potentiates the radical scavenging activity of *Ammi visnaga* L. when grown in drought-stress conditions [10]. In tomato plants, SA treatments obviously alleviate cadmium-induced growth inhibition by decreasing the cadmium accumulation and malondialdehyde (MDA) level as well as increasing CAT activity and chlorophyll content [11]. Treatments also regulate root cell-wall composition through nitric oxide signaling in rice (*Oryza sativa* L.) [12]. In maize, SA increases crop yield through the enhancement of photosynthesis and antioxidant capacity under salt-stress conditions [13]. Exogenous SA at low concentrations alleviates the accumulation of pesticides and mitigates pesticide-induced oxidative stress in cucumber plants (*Cucumis sativus* L.) by increasing biomass, chlorophyll and proline contents, as well as ascorbate peroxidase and glutathione S-transferase activities, and by decreasing the MDA and H_2_O_2_ contents [14].

Sweet potato, *I. batatas* (L.) Lam, is an important food crop in China, which has the largest planting area in the world. In 2018, the sweet potato planting area in China reached 2.374 million hectares, accounting for 29.1% of the world planting area. The total output of fresh potato was 62.347 million tons, accounting for 57.11% of the worldwide output. The average yield per hectare was 22.44 tons, the highest in the world [15]. As a significant crop, it provides calories, proteins, vitamins, edible fiber and minerals for humans. In the past, its consumption has saved millions of people from starvation, and it is presently popular with city dwellers in southeastern coastal China for obesity prevention and weight reduction. With economic development, many snack foods based on fresh and processed potatoes have been developed, and this has effectively increased farmer income and alleviated poverty. Therefore, sweet potato production has increased in China in recent years, especially in the southeast coastal areas [2]. As an upland crop, sweet potato is highly tolerant to water deficits. In the south of China, sweet potato is usually planted in hilly areas, where the farmlands are mostly rainfed, but in the north of China water deficits are very common [16]. A water deficiency usually results in a reduced crop yield. Thus, there is a need to develop new measures to alleviate the negative impacts of drought stress on sweet potato production.

Although SA can protect plants from biotic and abiotic stresses through varied metabolic mechanisms, the complete regulatory process is not clear. Moreover, there are limited reports regarding the effects of SA applications on sweet potato plant protection. In this study, we investigated the extent to which SA improves sweet potato tolerance in multiple manners, including seedling growth, chloroplast membrane protection, osmotic adjustment, oxidative stress, antioxidant balance and ABA-related gene expression levels under drought conditions.

## 2. Results and Analysis

### 2.1. SA Protects Photosynthetic Pigments and Increases the Photosynthetic Rate

Drought stress resulted in a significant decrease in the total chlorophyll content of sweet potato leaves and spraying appropriate concentrations of SA solutions on the leaves effectively reversed this adverse impact (Table 1). Chlorophyll a (Chl a) represents a portion of the total amount of chlorophyll and is a major component of the photosystem (PS) I. As shown in Figure 1a,b, with the extension of drought-stress time, the Chl a contents in the leaves of ‘Zheshu 77’ (‘ZS77’) and ‘Zheshu 13’ (‘ZS13’) showed downward trends.

For ‘ZS77’, there was no significant difference in the Chl a content among the treatments after 24 h. At 48 h, C1 and C2 were significantly increased by 21.68% and 17.25%, respectively, compared with C0. At 72 h, C2 and C3 were significantly increased by 20.10% and 21.16%, respectively, compared with C0. For ‘Zheshu13’, the C1, C2 and C3 treatments all alleviated the decline in the Chl a content, but the C4 treatment showed a greater decline than C0. The two sweet potato varieties showed limited differences in SA concentrations, but the concentrations used in the C2 and C3 treatments, 2.0 mg·L^−1^ and 4.0 mg·L^−1^, respectively, significantly alleviated the decline in the Chl a content in both varieties.

The drought treatment caused pronounced reductions in the chlorophyll b (Chl b) contents in both varieties. For ‘ZS77’, Chl b contents at 48 h and 72 h decreased by 17.46% and 22.22%, respectively, compared with 24 h after the C0 treatment (Figure 1c). The application of exogenous SA increased the Chl b content. The contents of Chl b after the C2 and C3 treatments significantly increased by 9.52% and 15.87%, respectively, 26.92% and 25.00%, respectively, and 28.57% and 26.53%, respectively, compared with after C0 at 24, 48 and 72 h, respectively. For ‘ZS13’, only the C3 treatment significantly increased the Chl b contents at 24, 48 and 72 h (Figure 1d).

For ‘ZS77’, at 24, 48 and 72 h after drought treatments, the net photosynthetic rate (Pn) decreased by 14.18%, 49.55% and 60.67%, respectively. The drought stress also resulted in considerable decreases in the transpiration rate (Tr) and leaf stomatal conductance (Gs) and an increase in the intracellular CO_2_ concentration (Ci) for both varieties after 24, 48 and 72 h of drought stress (Table 2).

Seedling leaves receiving different concentrations of exogenous SA demonstrated obvious increases in the Pn, especially after 24 h and 48 h. The Pn values at 48 h after C1, C2 and C3 treatments increased 45.88%, 62.52%, and 36.97%, respectively, for ‘ZS77’, and by 13.90%, 52.70% and 6.42%, respectively, for ‘ZS13’, when compared with C0 (Figure 2a,b). Applications of SA solutions also modulated the decrease in Tr. Under drought conditions, after 24, 48 and 72 h of treatments, all the plant Tr values increased by different amounts. After 48 h of drought stress, the Tr values of the C1, C2, C3 and C4 treatments for ‘ZS77’ were significantly increased by 39.27%, 49.57%, 34.31% and 16.74%, respectively, compared with C0. For ‘ZS13’, the Tr values of the four treatments significantly increased by 52.91%, 61.70%, 79.19% and 35.33%, respectively (Figure 2c,d).

Spraying SA on leaves mitigated the decline in Gs caused by drought for both varieties. In general, C2 was more efficacious. At 24, 48 and 72 h after treatments, the Gs values in the two varieties increased by 26.56%, 55.16% and 18.95% and by 31.19% 78.40% and 65.48%, respectively (Figure 2e,f). The SA application also modulated an adverse increase in Ci values. The Ci values after the C2 and C3 treatments decreased by 27.23% and 23.45%, respectively, after 48 h and by 32.81% and 23.33%, respectively, after 72 h compared with those of C0. The mitigation effects were significantly different (*p* < 0.05) (Figure 2g,h).

### 2.2. SA Reduces Oxidative Stress and Enhances Antioxidant Enzyme Activities

Drought stress led to strict increases in the H_2_O_2_ and MDA contents. For ‘ZS77’, at 24, 48 and 72 h after drought treatment, the H_2_O_2_ content increased by 32.50%, 74.96% and 95.90%, respectively, and the MDA content increased by 15.96%, 43.41% and 72.95%, respectively. There were similar changes for ‘ZS13’ (Table 3). SA applications significantly decreased the H_2_O_2_ and MDA contents at 24, 48 and 72 h after treatments (Figure 3 and Figure 4)

At 72 h after foliar spraying SA, the H_2_O_2_ contents of ‘ZS77’ receiving C1, C2 and C3 treatments significantly decreased by 18.20%, 31.65% and 26.95%, respectively, compared with C0, whereas the H_2_O_2_ contents of ‘ZS13’ significantly decreased by 17.50%, 33.60% and 30.05%, respectively (Figure 3). At 48 h after treatment with SA, the MDA contents in ‘ZS77’ receiving C1, C2 and C3 treatments decreased by 14.28%, 25.47% and 17.29%, respectively, compared with the C0, and they decreased by 18.77%, 37.99% and 29.17%, respectively, after 72 h. ‘ZS13’ plants receiving the C1, C2 and C3 treatments showed similar trends (Figure 4). The MDA content after the C4 treatment maintained the same upward trend as that of the C0, with no significant difference.

Under drought conditions, the SOD activity levels in ‘ZS77’ and ‘ZS13’ leaves increased along with time (Figure 5a,b). For ‘ZS77’, after 48 h and 72 h of drought stress, the SOD activity increased by 8.44% and 28.72%, respectively, compared with C0 after 24 h. The SA treatment further increased the SOD activity. At 24, 48 and 72 h after the C2 treatment the SOD activities increased by 34.90%, 37.63% and 42.31%, respectively, and after the C3 treatment they increased by 42.03%, 51.41% and 43.64%, respectively, compared with those of the C0 treatment. The C1 and C4 treatments also significantly increased the SOD activities, but to lesser degrees than C2 and C3. The changes in the SOD activities of ‘ZS13’ showed similar trends for all the treatments.

POD works in cooperation with SOD and CAT to scavenge ROS. With the extension of drought-stress treatment times, the POD activity levels in the two sweet potato variety leaves increased significantly, and SA applications promoted a further increase in the POD activities. The variety ‘ZS77’ showed a higher POD activity level independent of the SA treatments (Figure 5c,d). For ‘ZS77’, at 24, 48 and 72 h after the C2 treatment, the POD activities increased by 96.39%, 72.08% and 78.09%, respectively, compared with C0, whereas for ‘ZS13’, the POD activities increased by 39.35%, 64.38% and 73.58%, respectively. The differences between varieties were significant (*p* < 0.05).

CAT catalyzes the decomposition of H_2_O_2_ into H_2_O and O_2_ in vivo to help plants cope with drought stress. With the extension of the C0 drought-treatment time, the CAT activity levels in leaves of the two sweet potato varieties increased gradually. After spraying different concentrations of SA solutions on the leaves, the CAT activities showed varying degrees of a downward trend, and the resulting activity range positively correlated with the SA concentration (Figure 5e,f). For ‘ZS77’, the CAT activity levels decreased by 3.99%, 17.41%, 26.20% and 29.95% after C1–4 treatments, respectively, at 24 h compared with the C0 treatment, and at 48 h, the values had decreased by 24.67%, 36.43%, 47.95% and 49.50%, respectively. After 72 h, the levels were 34.75%, 43.62%, 47.36% and 54.40%, respectively, compared with C0. The ‘ZS13’ variety showed a similar trend.

### 2.3. SA Mediates the Plant Osmotic Status

Compared with normally growing plants, after 24, 48 and 72 h of drought stress, the leaf relative water content (RWC) values decreased by 9.16%, 12.48% and 17.48%, respectively. SA treatments, especially C2 and C3, mediated the RWC decreases at 24, 48 and 72 h. At 48 h after the SA treatment of ‘ZS77’, the RWC values of C2- and C3-treated plant leaves significantly increased by 4.83% and 3.97%, respectively, compared with C0, whereas for ‘ZS13’, the RWC values significantly increased by 10.39% and 10.04%, respectively (Figure 6a,b). 

At 24, 48 and 72 h after drought treatments, compared with normally growing plants, the soluble sugar contents increased by 42.99%, 80.84% and 81.21%, respectively. Spraying exogenous SA further increased the soluble sugar contents independent of the concentration (Figure 7a,b). The C2 and C3 treatments were the most efficacious. For ‘ZS77’ at 24, 48 and 72 h after treatment, C2 increased by 28.78%, 30.19% and 51.96%, respectively, and C3 increased by 17.28%, 24.79% and 55.93%, respectively, compared with C0. There was a similar trend in ‘ZS13’. After SA treatments, the soluble protein contents in leaves also increased (Figure 7c,d). For ‘ZS77’ after 24 h of drought, the soluble protein contents of the C1–4 treatments significantly increased by 8.22%, 27.03%, 32.22% and 21.51%, respectively, compared with C0. The content range increased further at 48 h and 72 h after treatment. Similar changes occurred in ‘ZS13’.

### 2.4. SA Effects on Abscisic Acid (ABA) Content and NCED3-Like Gene Express

The ABA content of a plant is closely correlated with cell metabolism and growth. Under drought-stress conditions, at 24 h the ABA contents in C0-treated leaves of ‘ZS77’ plant increased by 146.45% compared with CK leaves (Figure 8a). For ‘ZS77’, the ABA contents in leaves treated with C1–4 decreased by 26.18%, 40.98%, 34.74% and 26.99%, respectively, compared with C0. The C2 and C3 treatment effects were good. Similar changes occurred in ‘ZS13’.

As shown in Figure 9, the expression status of the *NCED3-like* gene in sweet potato showed an induced trend under drought-stress conditions, with the expression level increasing accordingly. However, the expression levels of the *NCED3-like* gene in sweet potato after exogenous SA treatments were lower than those of the C0, but still greater than those of the CK (normal water supply). The results of the two varieties were consistent.

### 2.5. SA Improves Growth Traits under Drought-Stress Conditions

Without drought stress, the two sweet potato varieties ‘ZS77’and ‘ZS13’ grow rapidly, with 2–3 cm of vine elongation per day. The vine and leaf growth were greatly reduced, as well as the dry matter accumulation, under drought conditions (Appendix A). The foliar spraying of SA mitigated the impacts of drought stress on sweet potato growth (Table 4). For ‘ZS77’, the vine length significantly increased 20.32%, 19.10% and 11.01% after C2, C3 and C4 treatments, respectively, compared with the C0 treatment. Similar results were obtained in ‘ZS13’.

The dry matter weight (DW) is closely correlated with vine and root growth. For ‘ZS13’, the DWs of plants receiving C1–4 treatments increased 8.64%, 21.36%, 20.57% and 7.12% compared with C0. The C2 and C3 treatments reached the significance level. The variation in characteristics and difference in the significance of the ‘ZS77’ DW were consistent with those of ‘ZS13’. The leaf area per plant (LA) showed noticeable increases in the two varieties. For ‘ZS77’ and ‘ZS13’ the LA values of the C2 treatment significantly increased 16.44% and 18.01%, whereas those of the C3 treatment increased 22.11% and 22.40% compared with C0. The LA values of treatments C1 and C4 increased but did not reach the significance level in either variety.

## 3. Discussion

The seedling stage of sweet potato is the key period for root and vine growth and is followed by the early part of the rooting and branching stage. Zhang [17] reported that drought during the rooting and branching stage has the greatest negative impact on the final sweet potato yield. Field experiments conducted by Li [18] also showed that drought in the early sweet potato growth periods have the greatest negative impacts on root development. Until now, most experiments investigating SA applications alleviating drought, high or low temperature and other stresses have been conducted under non-field conditions. Fan [19] reported that under field experimental conditions, spraying 0.1 mmol·L^−1^ SA at the flowering stage mediates the high temperature stress of wheat and that it is more effective than spraying at other growth stages. In this study, the application of a proper SA concentration to drought plants improved all the studied growth traits (Table 4), with the most efficacious treatments, C2 and C3, having 2–4 mg·L^−1^ SA. The improvement was attributed to the SA-mediated ameliorating impact under drought-stress conditions that results from complex biochemical and physiological machinery. Similarly, SA-mediated improvements in growth traits under drought-stress conditions have been reported in wheat crops [20,21]. After reviewing the literature, this experiment sprayed SA for 3 days (twice daily) consecutively. However, this application rate would increase the cost of farming. Therefore, it is necessary to explore effective ways to reduce the spraying times to alleviate drought stress during production.

The application of the appropriate dose of an SA solution can promote the growth of sweet potato seedlings under drought conditions. To reduce water transpiration, sweet potato, as with other plants, will close or partially close the stomata under drought conditions, and this leads to reductions in the Ci and Pn. In this study, the Gs and Tr values of leaves from two sweet potato varieties decreased along with drought time, where the Ci increased, indicating that under drought conditions, the inhibition of photosynthesis in sweet potato was not primarily due to stomatal-related factors [2]. However, after the C2 and C3 treatments, the Gs, Tr and Pn values of ‘ZS 77’ and ‘ZS13’ leaves increased compared with the C0 (0.0 mg·L^−1^). An increase in Gs is conducive to the photosynthetic system obtaining more CO_2_ and increasing the Pn. Additionally, SA modulates the decline in Pn, which may be related to its protection of the D1 protein in chloroplast PS II. Huang [8] found that SA protects the D1 protein in chloroplasts of *Dendrobium officinale* under low temperature-stress conditions. Wang [22] reported that SA maintains a higher level of D1 protein phosphorylation, enhances the light energy capture efficiency of the PS II reaction center under heat and high irradiance stress, and inhibits the decline in the maximum photochemical efficiency (Fv/Fm) of PS II in wheat crops. Additionally, SA may effectively alleviate the stress of active oxygen in plants, stimulate the antioxidant defense system of plants and protect the photosynthetic apparatus from being damaged by excessive active oxygen to ensure photosynthesis progresses.

ROS are collections of several types of active molecules, including superoxide anion (O_2_^−^), hydroxyl radical (^−^OH), hydroperoxyl radical (HO_2_^−^), an alkoxy radical (RO^−^), and non-radicals such as hydrogen peroxide (H_2_O_2_) and singlet oxygen (^1^O_2_), which are inevitable products of normal plant growth and metabolism [23,24]. When plants are exposed to biotic or abiotic stresses, a growth disorder normally occurs, such as membrane damage, ROS generation and excessive accumulation, which can lead a serious injury. To alleviate or prevent ROS-induced oxidative injury, plants have evolved mechanisms to scavenge these toxic and reactive species through the antioxidation of enzymatic and nonenzymatic systems [25]. Drought leads to excess electrons being captured by plant chloroplasts and in the production of ROS. ROS damages the membrane lipid systems of cells and leads to increased MDA levels [11]. Here, SA applications decreased the MDA content, which suggested that the exogenous SA protected the membrane lipid system. The declines in Chl a and b may be related to ROS over-accumulation. High levels of ROS destroy the chloroplast membrane, resulting in the chlorophyll synthesis of sweet potato being faster than the decomposition under drought conditions [2]. SA is involved in the regulation of plant antioxidant enzyme systems as a signal molecule. Recently, Ma [26] showed that exogenous SA treatments induce the upregulation of *PIP1* expression in *Lycium ruthenicu* leaves under drought conditions, increase the contents of plant plasma membrane intrinsic proteins, enhance the H_2_O_2_ transport capacities of plasma membranes, and help trigger the activities of SOD, POD and other antioxidant enzymes. Peng [27] reported that exogenous SA applications induce the over-expression of the SA-binding protein 2 gene, which plays active roles in regulating the expression of antioxidant enzyme-related genes and the activities of antioxidant enzymes in tobacco plants. In this study, foliar spraying of an appropriate SA dose increased the SOD and POD activities, which is consistent with these previous results. However, after SA applications, the CAT activity levels in all treatments decreased. This may be because exogenous SA specifically binds to CAT and inhibits its activity, thereby increasing the H_2_O_2_ contents of cells. H_2_O_2_, as the second messenger in cells, can activate the expression of corresponding resistance genes in plants and promote the responses of antioxidant defense systems in cells [28]. In agreement with our results, several other studies have suggested that the SA applications inhibit CAT activity in other plants [29,30,31], which suggests that the applications do not positively regulate CAT activity.

SA applications appeared to increase the RWC, which is very beneficial for maintaining normal physiological functions of leaves under drought conditions (Figure 6). A similar increase in leaf RWC was observed during drought stress in a wheat crop [32]. SA also increased osmolytes, including soluble sugar and protein, under drought conditions (Figure 7). The low-molecular antioxidants, such as ascorbic acid, glutathione, α-tocopherol, β-carotene, flavonoids and proline, are important soluble protein compounds and play vital roles in maintaining the reducing environment [4,20]. Here, the soluble carbohydrates cooperated with soluble proteins in maintaining cell turgor pressure and helped stabilize cell membranes, providing a stable internal environment for optimum metabolic and physiological activities under drought-stress conditions. Khalvandi et al. [33] found that SA applications on six ecotypes of wheat crops under drought-stress conditions cause marked increases in osmolytes (soluble carbohydrates and protein contents), which supports our results.

When plants are in a drought environment, the ABA contents of roots increase and are transported to the leaves to guide stomatal closure or partial closure. In this study, the ABA contents in leaves decreased after they were sprayed with SA. Furthermore, the expression of ABA synthesis-related *NECD-like3* decreased in leaves. The results suggest that SA regulates the ABA content, perhaps through the expression of *NECD-like3*. Thus, the metabolic pathway of exogenous SA acting on *NECD-like3* expression deserves further study. Zhang [34] reported that SA not only promotes seed dormancy and inhibits the transformation of seed from dormancy to germination, but it also produces a strong germination inhibition effect by increasing the sensitivity of seed to exogenous ABA. Whether SA enhances the sensitivity of sweet potato to endogenous ABA when applied for drought tolerance improvement deserves further study.

In conclusion, foliar spraying of an appropriate SA concentration enhanced the activity levels of SOD and POD and protected the photosynthetic apparatus of leaves. Then, the leaf Pn increased and plant growth indicators were upregulated in sweet potato under drought conditions. The SA downregulated the expression of ABA-related genes, such as *NECD-like3*, in sweet potato leaves and decreased the ABA content in leaves. These functions may be important for the enhancement of drought tolerance in sweet potato plants and; therefore, have practical application potentials.

## 4. Materials and Methods

### 4.1. Plant Materials and Growing Conditions

The sweet potato varieties ‘Zheshu 77’ (ZS77) with chicken claw leaves and ‘Zheshu 13’ (ZS13) with horseshoe leaves planted in Zhejiang Province were selected as experimental plants. The seedlings were transferred to plastic pots (top and bottom pot diameters of 16 cm and 12 cm, respectively; pot depth of 17 cm) containing 1.0 kg clay loam soil from the Zijingang Campus experimental field at the Agricultural Experiment Station of Zhejiang University (AES-ZJU). The soil pH was 6.7, and it contained 28.48 g·kg^−1^ total soil organic matter, 1.68 g·kg^−1^ total nitrogen, 62.1 mg·kg^−1^ available phosphorus and 52.4 mg·kg^−1^ available potassium. Every pot contained one seedling, which had three grown leaves and was 8 cm high at planting. The seedlings were grown in a greenhouse at the Zijingang Campus of AES-ZJU with a temperature regime of 25/20 °C day/night and natural sunlight before the drought treatment. The plants were irrigated once every 2 or 3 d to avoid water stress. After a new leaf was fully spread, healthy and uniform seedlings (with 4–4.5 leaves and 12-cm heights) were selected for experiments. The experiments were carried out at the AES-ZJU from April to June 2020. Owing to the COVID-19 pandemic, some laboratory tests were completed in 2021.

### 4.2. SA Treatment and Drought Stress

The sweet potato seedlings were cultured in three climate chambers (AGCM-113DC01, Hangzhou, China) at 25/20 °C with a 14-h/10-h light/dark regime, with a photosynthetic photon flux density of 360 μmol_·_m^−2^_·_s^−1^ and 80% ± 5% relative humidity for 3 d as a pretreatment. During the pretreatment, the pots were irrigated regularly to maintain the soil water content (SWC) at the field capacity, which is 35.21% (*w*/*w*) and used as the CK status. After the SWC pretreatment, the drought and SA treatment pots were adjusted to 30% of field capacity, at 10.5 ± 0.5% SWC. The detail methods were described as previously reported [2]. After the SWC pretreatment, the plots were maintained at drought and CK states, the seedlings underwent independent foliar spraying with one of four SA concentrations (1.00, 2.00, 4.00 and 8.00 mg·L^−1^, abbreviated as C1, C2, C3 and C4, respectively). There was also a distilled water control (C0) treatment. Plants in each pot were sprayed with 10 mL of a SA solution twice daily at 8 a.m. and 5 p.m. for 3 consecutive days (six times total). There were 30 trays of plants in every treatment set, with three replications of 10 trays per replication, and the trays were randomly arranged. The third and fourth leaves from the tops of the stems were sampled at 24, 48 and 72 h after treatments. Half of each replication plant set was transported back to the greenhouse and cultured for an additional 7 d at the original SWC of either the treatment or CK. Then, the agronomic traits were investigated. All the green leaves from the sample plants in each treatment were sampled, frozen in liquid nitrogen and stored at −70 °C.

### 4.3. Determination of the Agronomy Traits

The agronomy traits were analyzed on the end day and at 7 d after treatment. The leaf area was determined using LI-6400 portable photosynthesis equipment (LI-COR, Lincoln, Nebraska, USA). The vine length represents the length from the apical of the shoot to the pot soil surface. Five plants from every replication were taken and dried at 80 °C to determine the dry weights (DWs), and the averages of each agronomic trait used data from five plants.

### 4.4. Determination of Chlorophyll Content and Photosynthetic Parameters

The chlorophyll content was determined using the method proposed by Lichtenthaler [35] and described previously [4]. The photosynthetic parameters of net photosynthetic rate (Pn), leaf stomatal conductance (Gs), intracellular CO_2_ concentration (Ci) and transpiration rate (Tr) were determined in the treated plants using LI-6400 equipment. The air temperature, relative humidity, CO_2_ concentration and photosynthetic photon flux density were maintained at 25 °C, 85%, 380 µmol·mol^−1^ and 1000 µmol·m^−2^·s^−1^, respectively.

### 4.5. Determination of H_2_O_2_ and Malondialdehyde (MDA) Content

The H_2_O_2_ levels were measured by monitoring the absorbance at 410 nm of the titanium–peroxide complex following the method described by Lin et al. [36] and briefly described in a previous report [8]. The MDA level was determined using the thiobarbituric acid reaction as described by Wu et al. [37].

### 4.6. Determination of Antioxidant Enzyme Activities

The total SOD activity was determined as described by Prochazkova et al. [38] and as used in our previous studies [4,8]. The methods to determine CAT and POD activity levels have been described previously [2,4].

### 4.7. Determination of Relative Water Content (RWC) and Soluble Carbohydrate and Protein Contents

Similar or the same leaves after chlorophyll determination were used in the RWC assay. The fresh weights (FWs), FWs at full turgor (TWs) and DWs of sample leaves were measured to determine the RWC [RWC (%) = (FW − DW/TW − DW) × 100%]. The soluble sugar content was measured using the anthrone colorimetry method [39]. The soluble protein content was estimated using the Coomassie brilliant blue staining method [40].

### 4.8. Determination of the ABA Content and Semi-Quantitative RT-PCR Analysis of NCED3-Like Genes

The leaf ABA content was determined as described by Liu et al. [41] and analyzed using a Shanghai Jinkang ELISA kit (Shanghai Jinkang Bioengineering Co., Ltd., Shanghai, China) in accordance with the manufacturer’s instructions. The absorbance was recorded at 450 nm.

Total RNA extraction from treated and control plant leaves and cDNA preparation were performed as described by Lin et al. [42]. The RNA extractions were carried out using a TRIzol reagent kit (Invitrogen, Carlsbad, CA, USA) in accordance with the protocol provided by the manufacturer. The ReverTra Ace qPCR RT Kit (Toyobo, Osaka, Japan) was used for cDNA synthesis following the manufacturer’s protocol. Independent PCR reactions with equal amounts of cDNA were performed using *NCED3-like* and *β-actin* primers (Appendix A). The PCR conditions used were an initial denaturation at 94 °C for 3 min, followed by 35 cycles of denaturation at 94 °C for 15 s, annealing at 54 °C for 15 s and extension at 72° C for 1 min, with a final extension at 72 °C for 10 min. PCR products were resolved on a 1.2% (*w*/*v*) agarose gel for size verification.

### 4.9. Statistical Analyses

The statistical analysis was performed using a one-way analysis of variance. Comparisons between the treatment means were performed using a least significant difference test at the *p* ≤ 0.05 level.

## Figures and Tables

**Figure 1 ijms-23-14819-f001:**
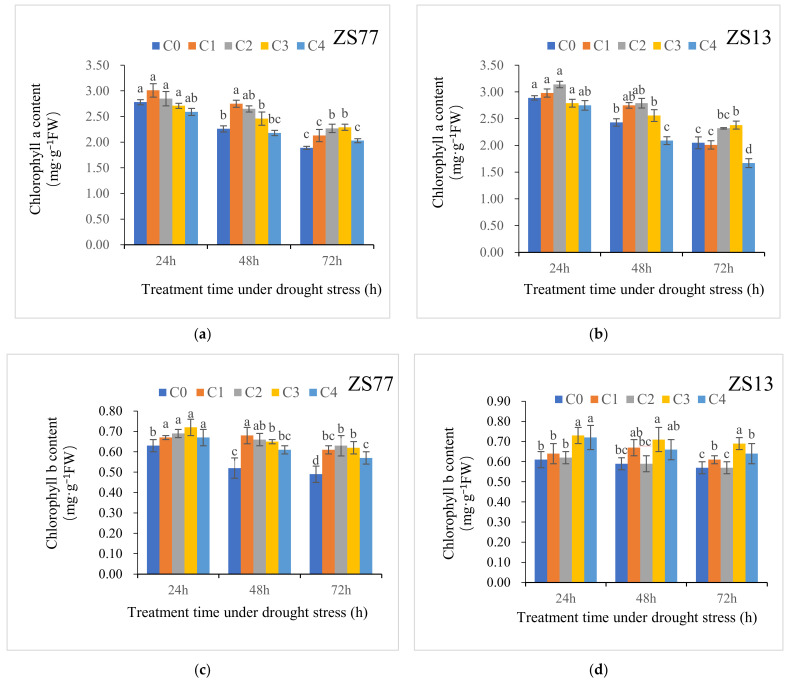
Effect of different concentrations of exogenous SA solution 0 (C0), 1.00 (C1), 2.00 (C2), 4.00 (C3), and 8.00 mg·L^−1^ (C4) on chlorophyll a (**a**,**b**) and b (**c**,**d**) content in sweet potato leaves under drought stress. ZS77 and ZS 13 refer to sweet potato varieties Zheshu 77 and Zheshu 13 respectively, the same as in following figures Bars with different letters indicate significant differences (*p* < 0.05).

**Figure 2 ijms-23-14819-f002:**
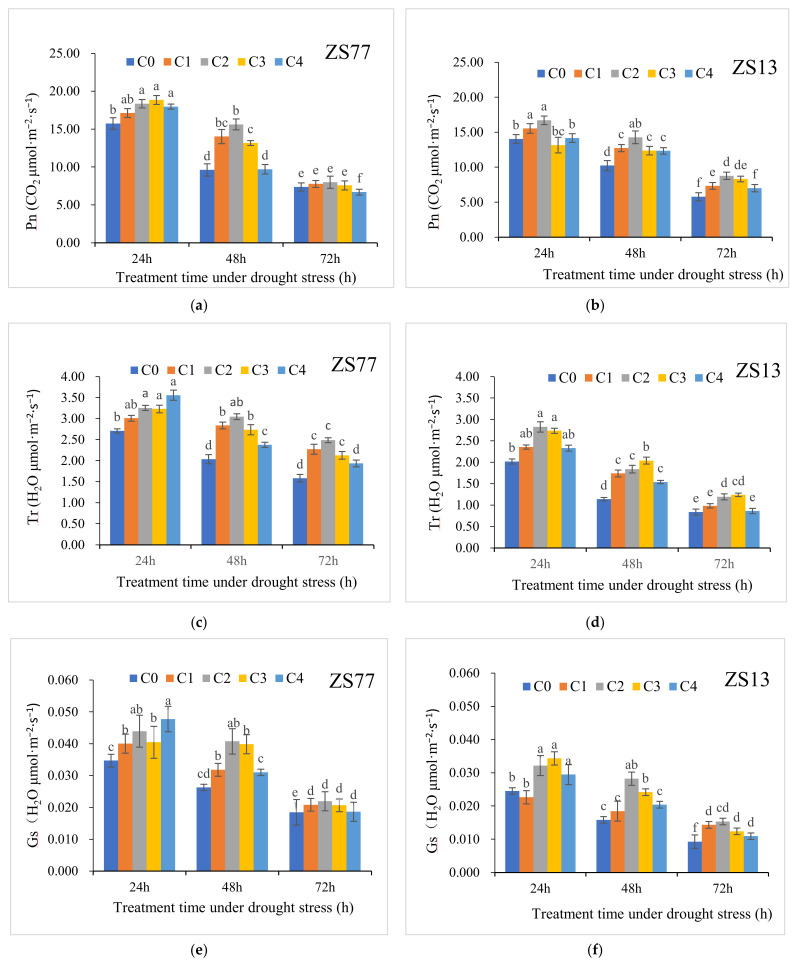
Effect of different concentrations of exogenous SA solution 0 (C0), 1.00 (C1), 2.00 (C2), 4.00 (C3), and 8.00 mg·L^−1^ (C4) on Pn (**a**,**b**), Tr (**c**,**d**), Gs (**e**,**f**), and Ci (**g**,**h**) in sweet-potato leaves under drought stress. Bars with different letters indicate significant differences among treatments at 0.05 level (Duncan).

**Figure 3 ijms-23-14819-f003:**
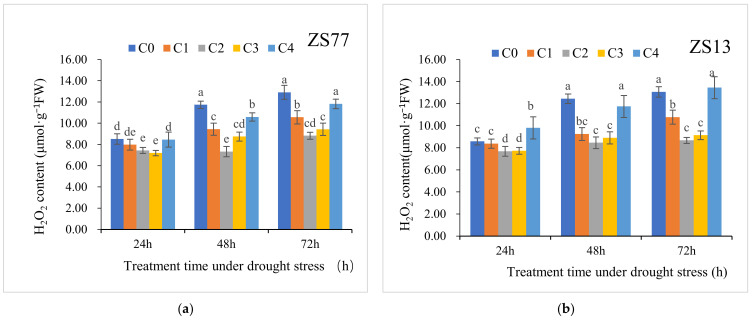
Effect of different concentrations of exogenous SA solution 0 (C0), 1.00 (C1), 2.00 (C2), 4.00 (C3), and 8.00 mg·L^−1^ (C4) on H_2_O_2_ content of sweet-potato leaves under drought stress. Bars with different letters indicate significant differences among treatments at 0.05 level (Duncan).

**Figure 4 ijms-23-14819-f004:**
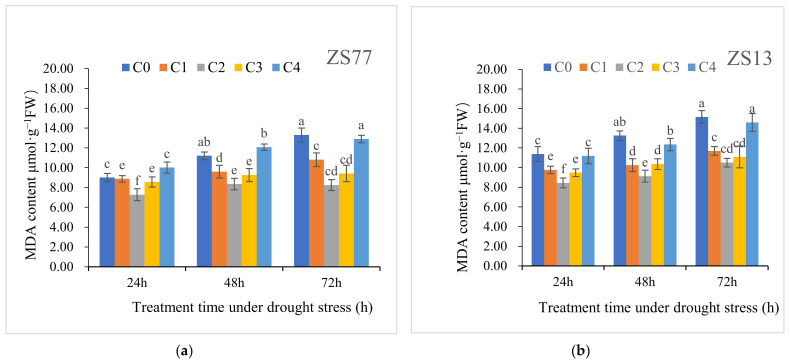
Effect of different concentrations of exogenous SA solution 0 (C0), 1.00 (C1), 2.00 (C2), 4.00 (C3), and 8.00 mg·L^−1^ (C4) on MDA content of sweet-potato leaves under drought stress. Bars with different letters indicate significant differences among treatments at 0.05 level (Duncan).

**Figure 5 ijms-23-14819-f005:**
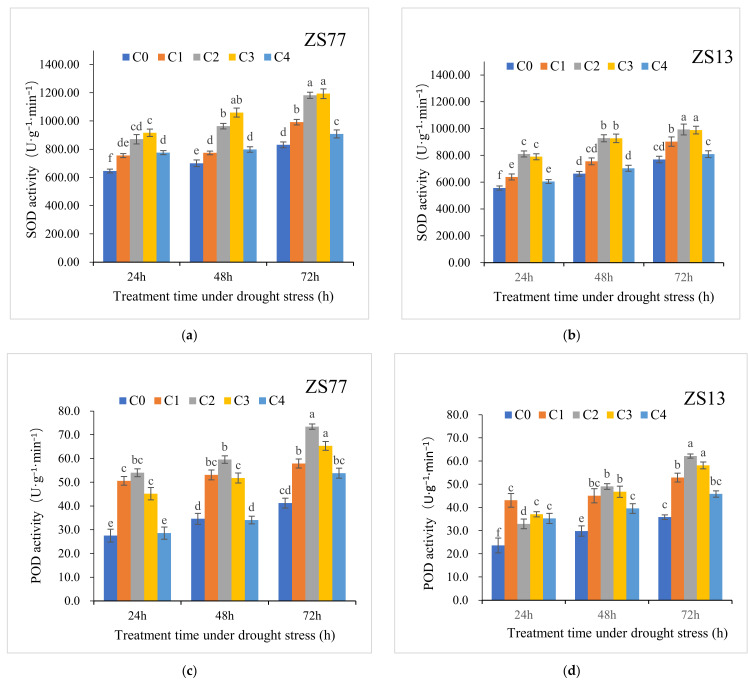
Effect of different concentrations of exogenous SA solution 0 (C0), 1.00 (C1), 2.00 (C2), 4.00 (C3), and 8.00 mg·L^−1^ (C4)on SOD (**a**,**b**), POD (**c**,**d**), CAT (**e**,**f**) activities in sweet-potato leaves under drought stress. Bars with different letters indicate significant differences among treatments at 0.05 level (Duncan).

**Figure 6 ijms-23-14819-f006:**
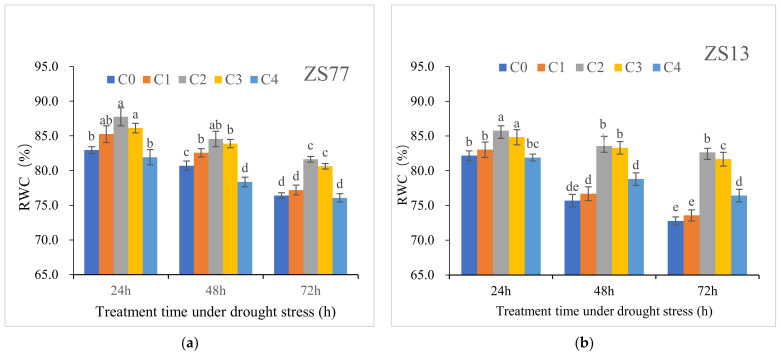
Effect of different concentrations of exogenous SA solution 0 (C0), 1.00 (C1), 2.00 (C2), 4.00 (C3), and 8.00 mg·L^−1^ (C4) on RWC of sweet-potato leaves under drought stress. Bars with different letters indicate significant differences among treatments at 0.05 level (Duncan).

**Figure 7 ijms-23-14819-f007:**
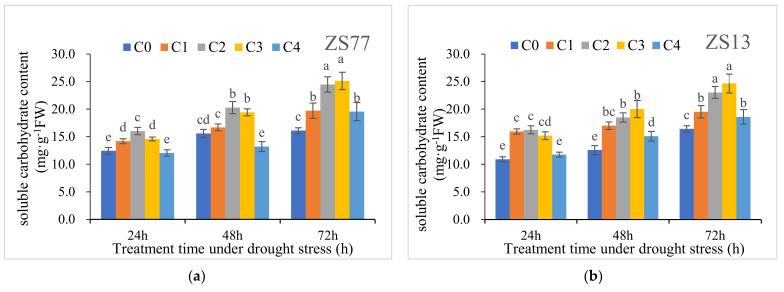
Effect of different concentrations of exogenous SA solution 0 (C0), 1.00 (C1), 2.00 (C2), 4.00 (C3), and 8.00 mg·L^−1^ (C4) on soluble sugar (**a**,**b**) and protein (**c**,**d**) in sweet-potato leaves under drought stress. Bars with different letters indicate significant differences among treatments at 0.05 level (Duncan).

**Figure 8 ijms-23-14819-f008:**
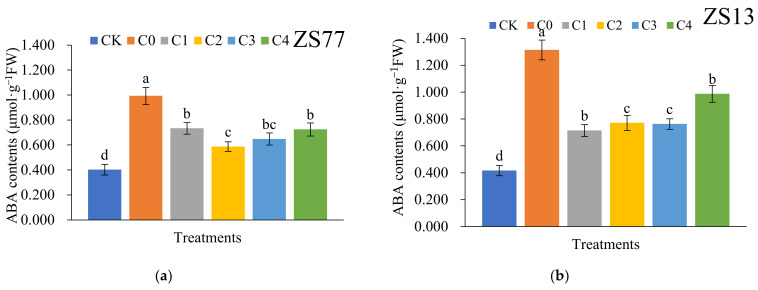
Effect of different concentrations of exogenous SA solution 0 (C0), 1.00 (C1), 2.00 (C2), 4.00 (C3), and 8.00 mg·L^−1^ (C4) on ABA content of sweet-potato leaves under drought stress. CK is the treatment with normal irrigation. Bars with different letters indicate significant differences among treatments at 0.05 level (Duncan).

**Figure 9 ijms-23-14819-f009:**
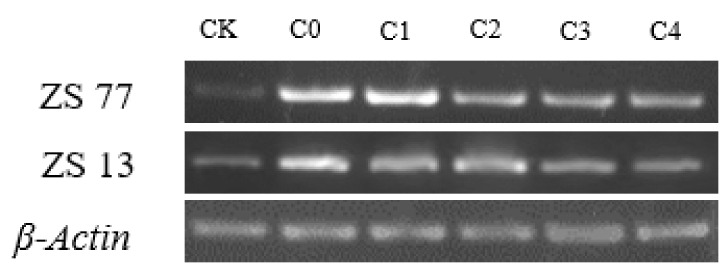
Semi-quantitative RT-PCR of *NCED3-like* gene. RNA was isolated from the leaves of normal irrigated (CK) and drought treated sweet potato plants foliar spraying 0 (C0), 1.00 (C1), 2.00 (C2), 4.00 (C3), and 8.00 mg·L^−1^ (C4) SA solution after 24 h treatment. Actin was used as an internal control. The experiment was repeated twice with the similar results.

**Table 1 ijms-23-14819-t001:** Effects of exogenous SA on total chlorophyll content of sweet potato under drought stress.

		Total Chlorophyll Content (mg·g^−1^FW)				
Cultivars	Treatment Duration (h)	C0	C1	C2	C3	C4
		0.0	1.00	2.00	4.00	8.00
ZS77	24	3.41 ± 0.08 b	3.68 ± 0.14 a	3.54 ± 0.16 a	3.43 ± 0.09 ab	3.26 ± 0.11 b
	48	2.78 ± 0.11 d	3.43 ± 0.11 ab	3.31 ± 0.09 b	3.11 ± 0.14 c	2.79 ± 0.07 d
	72	2.38 ± 0.07 e	2.74 ± 0.14 d	2.92 ± 0.13 c	2.91 ± 0.09 c	2.60 ± 0.07 d
ZS13	24	3.50 ± 0.08 b	3.62 ± 0.13 a	3.76 ± 0.09 a	3.52 ± 0.11 b	3.47 ± 0.15 b
	48	3.02 ± 0.10 c	3.42 ± 0.09 b	3.38 ± 0.13 b	3.27 ± 0.17 c	2.75 ± 0.12 d
	72	2.62 ± 0.14 d	2.62 ± 0.09 d	2.89 ± 0.08 c	3.07 ± 0.11 c	2.31 ± 0.13 e

The different lowercase letters show significant differences (*p* < 0.05).

**Table 2 ijms-23-14819-t002:** Effects of drought stress on photosynthetic characteristics of sweet potato leaves.

Item	Treatments	Duration of Drought Stress (h)					
		ZS77			ZS13		
		24 h	48 h	72 h	24 h	48 h	72 h
Pn (CO_2_ μmol·m^−2^ s^−1^)	CK	18.34 ± 0.57	19.05 ± 0.82	18.74 ± 0.77	16.58 ± 0.92	16.37 ± 1.11	17.05 ± 0.98
	C0	15.74 ± 0.77	9.61 ± 0.82	7.37 ± 0.54	14.04 ± 0.62	10.24 ± 0.73	5.78 ± 0.59
Tr (H_2_O mmol·m^−2^ s^−1^)	CK	5.38 ± 0.10	5.62 ± 0.09	5.55 ± 0.07	3.77 ± 0.08	3.69 ± 0.11	3.73 ± 0.09
	C0	2.71 ± 0.05	2.04 ± 0.11	1.58 ± 0.09	2.02 ± 0.06	1.14 ± 0.04	0.84 ± 0.07
Gs (H_2_O mol·m^−2^ s^−1^)	CK	0.214 ± 0.007	0.228 ± 0.005	0.237 ± 0.08	0.188 ± 0.04	0.170 ± 0.07	0.183 ± 0.10
	C0	0.035 ± 0.002	0.026 ± 0.001	0.018 ± 0.004	0.025 ± 0.001	0.016 ± 0.001	0.009 ± 0.002
Ci (CO_2_ μmol·mol^−1^)	CK	250.4 ± 4.8	256.8 ± 8.5	255.6 ± 10.1	223.9 ± 7.3	240.3 ± 5.2	239.2 ± 6.7
	C0	143.4 ± 4.6	190.6 ± 7.9	223.7 ± 8.1	155.6 ± 6.4	210.4 ± 8.0	244.3 ± 9.0

**Table 3 ijms-23-14819-t003:** Effects of drought stress on H_2_O_2_, MDA contents in sweet potato leaves.

Item	Treatments	Duration of Drought Stress					
		ZS77			ZS13		
		24 h	48 h	72 h	24 h	48 h	72 h
H_2_O_2_ content (μmol·g^−1^ FW)	CK	6.43 ± 0.38	6.71 ± 0.44	6.59 ± 0.51	7.55 ± 0.49	7.63 ± 0.33	7.59 ± 0.50
	C0	8.52 ± 0.49	11.74 ± 0.35	12.91 ± 0.66	8.58 ± 0.32	12.45 ± 0.43	13.07 ± 0.47
MDA content (μmol·g^−1^ FW)	CK	7.77 ± 0.31	7.81 ± 0.47	7.69 ± 0.62	8.40 ± 0.69	8.77 ± 0.63	8.64 ± 0.29
	C0	9.01 ± 0.41	11.20 ± 0.39	13.30 ± 0.70	11.38 ± 0.77	13.26 ± 0.48	15.16 ± 0.65

**Table 4 ijms-23-14819-t004:** Effects of exogenous SA on growth traits of sweet potato under drought stress.

	Vine Length (cm)		Dry Matter Weight (g)		Leaf Area (cm^2^)	
	ZS77	ZS13	ZS77	ZS13	ZS77	ZS13
C0	14.81 ± 1.67 b	15.46 ± 1.42 b	15.35 ± 1.78 b	16.43 ± 1.03 b	19.58 ± 1.35 c	20.71 ± 1.22 c
C1	15.62 ± 0.64 b	16.74 ± 0.67 ab	16.94 ± 2.37 ab	17.85 ± 2.78 ab	20.66 ± 1.67 cb	21.95 ± 1.54 bc
C2	17.82 ± 0.55 a	18.08 ± 0.85 a	19.36 ± 1.08 a	19.94 ± 1.47 a	22.82 ± 0.89 ba	24.44 ± 1.04 ab
C3	17.64 ± 0.61 a	18.18 ± 0.65 a	18.93 ± 0.78 a	19.81 ± 0.95 a	23.91 ± 0.89 a	25.35 ± 1.08 a
C4	16.44 ± 0.59 ab	16.76 ± 0.58 ab	17.75 ± 1.57 ab	17.6 ± 1.63 ab	21.41 ± 2.41 abc	22.45 ± 2.47 abc

The dry matter weight and leaf area were given for per plant, and the different lowercase letters show significant differences (*p* < 0.05).

## Data Availability

The raw data supporting the conclusions of this article will be made available by the authors without undue reservation.

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
