# Peer review of "Salicylic Acid Protects Sweet Potato Seedlings from Drought Stress by Mediating Abscisic Acid-Related Gene Expression and Enhancing the Antioxidant Defense System"

_ijms, 2022, doi:10.3390/ijms232314819_

Round 1
Reviewer 1 Report
In this manuscript, the authors mainly presented the function of SA in sweet potato seedlings drought tolerance. The physiological data indicated that the appropriate dose of SA should increase the drought resistance ability of sweet potato. The study has been well performed and presented; I have few points that might be addressed.
1, For the plant materials, why selected two different varieties. Please make an explanation.
2, The hormone ABA was considered to be the positive regulator to enhance plant drought resistance. Why the SA treatment decreased the ABA content under drought condition?
3, Figure legends are vague and lacks necessary explanations. Readers should be able to understand and grasp the main messages of each figure without reading the main text. Some figures also need to be revised to keep the standardized format.
Author Response
Dear editors and reviewers,
We are very glad to hear the information from you that our submission “Salicylic acid protects sweet potato seedlings from drought stress by mediating abscisic acid-related gene expression and enhancing the antioxidant defense system” may be accepted for publication in IJMS. We sincerely thank you and the two reviewers give us so great suggestions would make the manuscript perfect. About the comments we have done related reversion in new version of our manuscript. The revised sentences checked by our English as mother language colleague.
For convenience, we repeated each comment followed by our response.
Thank you.
Huang Chongping
For reviewer expert 1,
The responses to the comments are listed as bellow:
C1, For the plant materials, why selected two different varieties. Please make an explanation.
Response: We explained in line 374 and 375 in the paragraph Materials and Methods as “The sweet potato varieties ‘Zheshu 77’ (ZS77) with chicken claw leaves and ‘Zheshu 13’ (ZS13) with horseshoe leaves”. The leaf shape of chicken claw and horseshoe is two typical sweet potato leaf shape. These two varieties are popular in Zhejiang Province field production.
Both farmers and agronomists believe that chicken claw type leaf sweet potatoes are more drought resistant. But ‘Zheshu 13’ has higher yield and stronger growth capacity. This may require another article to study and demonstrate the problem.
C2, The hormone ABA was considered to be the positive regulator to enhance plant drought resistance. Why the SA treatment decreased the ABA content under drought condition?
Response: Yes, the hormone ABA was considered to be the positive regulator to enhance plant drought resistance. In this study, we find that the SA application down-regulated the NECD-like3 expression and may reduce the ABA synthesis and decreased the ABA content in sweet potato leaves. The SA application reduced ABA content in plant leaves, then increased the Gs (line 151 to 157) and Pn, thus the SA promoted the sweet potato growth (line 305,306). In paragraph discussion, line 354 to 364 we discussed the relationships among the SA application, NECD-like3 expression, leaf ABA content. However, we also consider this phenomenon deserves further study and mentioned in discussion. We agree with reviewers’ opinion and we may find a little scientific evidence and need more deep research.
C3, Figure legends are vague and lacks necessary explanations. Readers should be able to understand and grasp the main messages of each figure without reading the main text. Some figures also need to be revised to keep the standardized format.
Response: Yes, all Figure legends are revised and some figures are revised to keep the standardized format. The revised figure legends checked by our English as mother language colleague.
Thank you!
Huang Chongping
Reviewer 2 Report
The paper by Chongping Huang, Junlin Liao, Wenjie Huang and Nannan Qin, is about salicylic acid (SA)-mediated defence mechanisms under drought conditions in two sweet potato varieties. According to authors SA can protect plants from different stresses through varied metabolic mechanisms but the overall picture is not fully understood. Besides, there are limited reports on the effects of SA applications on sweet potato plant protection.
Authors investigated the extent to which SA improves sweet potato tolerance in many ways, among others they checked seedlings growth, chloroplast membrane protection, osmotic adjustment, oxidative stress, antioxidant balance and ABA-related gene expression levels under drought stress.
In my opinion the manuscript is quite well written. Research topic corresponds to the title. I have doubts about the results because nowhere in any figure do we have a control or a baseline. We observe the results from 24 hours. Besides, I have doubts about the drawings. The figures are not very aesthetically pleasing. They vary in size, borders are sometimes there and sometimes they are not. In Figure 8 even the scale has changed.
It would be good to fix them because it gives the impression of sloppiness.
In my opinion, this manuscript can be recommended for a publication after taking into account the above doubts.
Author Response
Respected reviewer expert, We have been engaged in sweet potato crop research and new technology extension for more than 7 years, and we feel that the drought has always been a big trouble to be solved in sweet potato production. However, there has been no systematic report on the study of salicylic acid to alleviate drought stress for sweet potato whether in Chinese or English. We hope that this study has theoretical and practical significance.
In the experimental design, we did not set the sampling time point immediately after the treatments. Because we felt that sampling 24, 48 and 72 hours after the completion of treatment might more comprehensively for investigation the various metabolic characteristics. Therefore, we take the C0 treatment at each sampling time point, that is, the treatment without spraying salicylic acid, as the control and baseline. Since in experimental design the CK is defined for regular irrigation, thus in most cases we analyzed the results in compare the data of different concentration SA solution (C1-C4) with C0(without SA application). We hope that this is reasonable and can explain the results.
Due to we did not use the MDPI template in the initial submission, the layout of Figure 8 is irregular. We made adjustments in the revision. We hope the revised the manuscript is perfect.Thank you very much for your kind comments!
Huang Chongping